# Somatostatin, Cortistatin and Their Receptors Exert Antitumor Actions in Androgen-Independent Prostate Cancer Cells: Critical Role of Endogenous Cortistatin

**DOI:** 10.3390/ijms232113003

**Published:** 2022-10-27

**Authors:** Prudencio Sáez-Martínez, Francisco Porcel-Pastrana, Jesús M. Pérez-Gómez, Sergio Pedraza-Arévalo, Enrique Gómez-Gómez, Juan M. Jiménez-Vacas, Manuel D. Gahete, Raúl M. Luque

**Affiliations:** 1Maimonides Biomedical Research Institute of Cordoba (IMIBIC), 14004 Cordoba, Spain; 2Department of Cell Biology, Physiology, and Immunology, University of Cordoba, 14004 Cordoba, Spain; 3Reina Sofia University Hospital (HURS), 14004 Cordoba, Spain; 4CIBER Physiopathology of Obesity and Nutrition (CIBERobn), 14004 Cordoba, Spain; 5Urology Service, Reina Sofia University Hospital, 14004 Cordoba, Spain

**Keywords:** somatostatin, cortistatin, prostate cancer, somatostatin analogues, therapeutic tool

## Abstract

Somatostatin (SST), cortistatin (CORT), and their receptors (SSTR1-5/sst5TMD4-TMD5) comprise a multifactorial hormonal system involved in the regulation of numerous pathophysiological processes. Certain components of this system are dysregulated and play critical roles in the development/progression of different endocrine-related cancers. However, the presence and therapeutic role of this regulatory system in prostate cancer (PCa) remain poorly explored. Accordingly, we performed functional (proliferation/migration/colonies-formation) and mechanistic (Western-blot/qPCR/microfluidic-based qPCR-array) assays in response to SST and CORT treatments and CORT-silencing (using specific siRNA) in different PCa cell models [androgen-dependent (AD): LNCaP; androgen-independent (AI)/castration-resistant PCa (CRPC): 22Rv1 and PC-3], and/or in the normal-like prostate cell-line RWPE-1. Moreover, the expression of SST/CORT system components was analyzed in PCa samples from two different patient cohorts [internal (*n* = 69); external (Grasso, *n* = 88)]. SST and CORT treatment inhibited key functional/aggressiveness parameters only in AI-PCa cells. Mechanistically, antitumor capacity of SST/CORT was associated with the modulation of oncogenic signaling pathways (AKT/JNK), and with the significant down-regulation of critical genes involved in proliferation/migration and PCa-aggressiveness (e.g., *MKI67*/*MMP9*/*EGF*). Interestingly, CORT was highly expressed, while SST was not detected, in all prostate cell-lines analyzed. Consistently, endogenous CORT was overexpressed in PCa samples (compared with benign-prostatic-hyperplasia) and correlated with key clinical (i.e., metastasis) and molecular (i.e., *SSTR2*/*SSTR5* expression) parameters. Remarkably, CORT-silencing drastically enhanced proliferation rate and blunted the antitumor activity of SST-analogues (octreotide/pasireotide) in AI-PCa cells. Altogether, we provide evidence that SST/CORT system and SST-analogues could represent a potential therapeutic option for PCa, especially for CRPC, and that endogenous CORT could act as an autocrine/paracrine regulator of PCa progression.

## 1. Introduction

Prostate cancer (PCa) represents the most common cancer type among men worldwide and the second cause of cancer-related death in this collective [1]. The main problem associated with this pathology is the management of the advanced disease, which consequently represents more than 80% of PCa-related deaths [2]. Due to its hormone dependency, the diagnosis of advanced PCa is followed by androgen deprivation therapy (ADT), usually combined with certain drugs such as Abiraterone or Enzalutamide/Daroluamide/Apalumatied, which block the androgen receptor signaling pathway in PCa cells [3,4]. Unfortunately, between 10–20% of the patients develop resistance to these approaches, leading to the development of the most aggressive PCa phenotype, called castration-resistant prostate cancer (CRPC) [5,6]. Currently, this stage is mainly treated with androgen-synthesis inhibitors (e.g., Abiraterone), androgen receptor (AR)-inhibitors (e.g., Enzalutamide), and/or taxanes (e.g., Docetaxel) [7]. In addition, new therapies are being introduced to manage the advanced stage of PCa including platinum-based therapies (e.g., Cisplatin, carboplatin, etc.) or PARP-inhibitors (e.g., Olaparib, Rucaparib, etc.) [8,9,10,11]. However, despite the improvement in the overall survival associated with the aforementioned therapies, CRPC remains lethal nowadays [5,6]. Therefore, it is necessary to further understand the biology of PCa, in order to develop novel or optimize available medical therapeutic approaches to tackle this disease, especially the CRPC phenotype.

In this sense, the somatostatin, cortistatin and somatostatin-receptors (SST/CORT/SSTRs) system represents a useful source of diagnostic/prognostic biomarkers and therapeutic targets to manage and treat various endocrine-related cancers (ERCs), owing to its pleiotropic functional role encompassing whole body homeostasis to cancer cell functioning in different tumor types, wherein this system commonly acts to inhibit multiple processes, such as hormone secretion and cell proliferation, migration and invasion [12,13,14,15,16,17]. In fact, we have recently demonstrated that certain components of the SST system, especially some SSTR-subtypes [SSTR1-5, encoded by the somatostatin receptor 1–5 genes (*SSTR1-5*)], are dysregulated in PCa tissues and cells, wherein they play a relevant role in the pathophysiology of this disease [18,19,20]. Specifically, the presence of SSTR1 and the truncated splicing variant sst5TMD4 could exert relevant pathophysiological roles by regulating different tumor parameters in PCa cells [e.g., cell proliferation, migration and PSA secretion; disruption of the normal response to somatostatin analogs (SSAs)] [18,19,20]. However, the presence and/or functional roles of other key components of this hormonal system, including the endogenous ligands SST and CORT (both able to bind all SSTRs with comparable affinities [21]), have hitherto not been fully explored in PCa. Moreover, due to the relevance of this hormonal system in cancer, synthetic SSAs [e.g., first generation (octreotide, lanreotide), and second generation (pasireotide)] have been developed and are widely used as valuable tools to treat multiple ERCs, including pituitary and neuroendocrine tumors [12,22,23]. However, attempts to apply SSAs in PCa have yielded controversial results since the limited studies reported so far did not show improvement in overall survival [24], and the mechanistic reasons for those clinical failures are not fully known.

Based on the information described above, the current study was aimed at exploring, for the first time, the presence of the entire SST/CORT/SSTRs system in PCa, and to perform a parallel comparison of the in vitro effects of SST and CORT peptides on different normal-prostate and prostate tumor (PCa and CRPC) cell models, in order to design new diagnostic, prognostic, and therapeutic approaches that could impact the management of PCa, especially CRPC.

## 2. Results

### 2.1. SST and CORT Treatment Exert Antitumor Actions Exclusively in Androgen-Independent PCa Cells, but Not in Androgen-Dependent PCa Cells or Normal Prostate Cells

Treatments with SST or CORT peptides [10^−7^ M; dose based on previous reports (see Material and Methods sections below)] did not alter the proliferation rate in normal-prostate (RWPE-1) or AD-PCa (LNCaP) cell models (Figure 1A); however, they significantly decreased proliferation rate in two AI-PCa cell models (22Rv1 and PC-3; Figure 1A). Additionally, SST treatment considerably decreased the number of colonies formed in 22Rv1 and PC-3 cells (Figure 1B, top-panel), while this inhibition was also significantly observed in response to CORT in 22Rv1, but not in PC-3, cells (Figure 1B, bottom-panel). Similarly, SST treatment significantly reduced the migration rate in PC-3 cells, while CORT only tended to reduce (*p* = 0.09) this capacity (Figure 1C).

### 2.2. SST and CORT Treatment Modulates the Levels of Key Oncogenic Signaling Pathways and Tumor-Related Genes in Androgen-Independent PCa Cells

Based on the results previously showed, we next explored the potential signaling-pathways modulated in response to SST and CORT treatment in AI-PCa cells. Firstly, phosphorylation levels of key proteins belonging to different oncogenic signaling pathways and/or associated with PCa development/aggressiveness (i.e., Protein kinase B (AKT), extracellular signal-regulated kinase (ERK), c-Jun N-terminal kinase (JNK), Phosphatase and Tensin Homolog (PTEN) and Androgen Receptor (AR)] were determined by Western blotting after 30 min of SST and CORT exposition (Figure 2A,B).

This analysis revealed that treatment with SST and CORT cells significantly decreased the phosphorylation levels of AKT, but not of ERK, PTEN or AR, in 22Rv1 and PC-3 cells (Figure 2A,B, respectively). Moreover, CORT (but not SST) also decreased the phosphorylation levels of JNK in 22Rv1 and PC-3 cells (Figure 2A,B, respectively). As previously reported elsewhere [25], PC-3 did not express PTEN nor AR at a protein level (Figure 2B).

To further understand the molecular mechanisms underlying the SST and CORT effects, we also evaluate the expression levels of key genes related to proliferation/cell-cycle, migration, and aggressiveness in AI-PCa models after 24 h of SST and CORT exposition (Figure 2C). In 22Rv1 cells, SST treatment significantly decreased the expression levels of the cyclin-dependent kinase 4 (*CDK4*), the cyclin-dependent kinase inhibitor D (*CDKND*), the matrix metallopeptidase 9 (*MMP9*), and the enhancer of Zeste homolog 2 (*EZH2*) (Figure 2C, left-panel). Moreover, SST treatment increased the expression levels of the cyclin-dependent kinase inhibitor 1A and 1B (*CDKN1A* and *CDKN1B*) and *PTEN* in 22Rv1 cells (Figure 2C, left-panel). Likewise, CORT treatment also decreased in 22Rv1 cells the expression levels of *MKI67*, N-Cadherin 2 (*CDH2*), *EGF*, and Proto-Oncogene C-Myc (*MYC*) (Figure 2C, right-panel); and increased the expression of *CDKN1A*, *CDKN1B*, and *CDKND* (Figure 2C, right-panel).

Similarly, SST and/or CORT treatment significantly reduced in PC-3 cells the expression levels of the proliferation markers *MKI67*, *CDK2*, *CDK4*, *CDK6*, *MMP3* (only CORT), *MMP9* (only SST), *MMP10* (only SST), the endothelial grow factor (*EGF*), *EZH2*, *MYC*, and the vascular endothelial growth factor receptor (*VEGFR*) (Figure 2C).

### 2.3. Expression of Somatostatin Receptors in Androgen-Independent PCa Cells and in PCa Tissues

We next interrogated the expression of all SSTR-subtypes in AI-PCa cells in order to identify which receptors might be mediating the antitumor actions and molecular-related events previously observed (Figure 1 and Figure 2) in response to SST and CORT treatment. A variable expression level for each of the SSTR-subtypes was found in 22Rv1 and PC-3 (Figure 3A). Specifically, the present work revealed that *SSTR1* and *SSTR5* are the dominant SSTR-subtypes expressed in 22Rv1 cells (mean ± SEM: 1.751 ± 592.9, and 2.172 ± 856.9 mRNA copy number, respectively), followed by significant lower levels of *SSTR2* > *sst5TMD4* > *SSTR3* (345.3 ± 90.65; 24.99 ± 7.902; 11.74 ± 5.84; respectively; *SSTR4* and *sst5TMD5* expression levels were very low or undetectable) (Figure 3A).

In PC-3 cells, *SSTR5* is the dominant SSTR-subtype expressed (mean ± SEM: 4,743 ± 1,893 mRNA copy number), followed by significant lower levels of *SSTR2* > *SSTR1* > *sst5TMD4* (46.70 ± 12.69; 14.12 ± 5.44; 13.17 ± 4.48, respectively, *SSTR3*, *SSTR4* and *sst5TMD5* expression levels were very low or undetectable) (Figure 3A).

Moreover, we also explored which SSTR-subtypes are expressed in human PCa tissues using the available samples from cohort-1 (Figure 3B). Specifically, we found that *SSTR1*, *SSTR2* and *SSTR5* were highly expressed in human PCa tissues (*SSTR1* ≥ *SSTR2* = *SSTR5*; mean ± SEM: 3,422,432 ± 362,369; 846,092 ± 110,588; and 602,159 ± 108,931 mRNA copy number, respectively). In contrast, expression levels of *sst5TMD4* were low, and *SSTR3*, *SSTR4* and *sst5TMD5* levels were very low or undetectable.

When viewing the results of Figure 3A,B together, it might be suggested that: (1) 22Rv1 and PC-3 were appropriate PCa cell models to perform the functional assays presented in this study (i.e., similar expression profile between AI-PCa cell models and human PCa tissues); and (2) human PCa might be sensitive to the actions of SST and CORT peptides as well as to different SSAs [first generation (octreotide; with high-affinity binding to SSTR2 and SSTR5) but specially to second generation (Pasireotide; a multireceptor-targeted SST with high affinity for SSTR1, SSTR2, SSTR3, and SSTR5)].

Finally, we also sought to determine whether endogenous *SST* and *CORT* were expressed in AI-PCa cells (Figure 3C). Interestingly, we found that endogenous *CORT*, but not *SST*, was highly expressed in 22Rv1 and PC-3 cells [mean ± SEM: CORT (2690 ± 1,595 and 1905 ± 889.9) vs. SST (44.12 ± 10.55 and 33.86 ± 16.21) mRNA copy number in 22Rv1 and PC-3, respectively; Figure 3C].

### 2.4. CORT Is Overexpressed in Human PCa Samples and It Is Associated with Aggressive Features

Based on the previous results, we next examined whether PCa tissues also express high levels of endogenous *CORT*. Our results revealed that, similar to the AI-PCa cell models previously analyzed (Figure 3C), *CORT* was also highly expressed in human PCa tissues. In fact, we demonstrated that *CORT* expression was significantly higher in PCa samples compared with BPH samples (used as controls; Figure 4A; cohort-1: see Materials and Methods below). Moreover, this differential expression was corroborated by Receiver Operative Characteristic (ROC) analyses since *CORT expression* levels was able to significantly discriminate between PCa vs. BPH samples, with an AUC (area under the curve) of 0.988 *(p* = 0.0045; Figure 4A).

Interestingly, although we did not observe any difference in the expression levels of endogenous *CORT* between primary tumors obtained from patients with metastasis compared to those without metastasis (cohort-1; Figure 4B), the expression of CORT was positively correlated with the expression of *SSTR2* and *SSTR5* in primary tumors obtained from patients with metastasis but not in those without metastasis [Figure 4C; a trend for significant (*p* = 0.1) was also observed for *SSTR1*]. Strikingly, analysis from the available Grasso in silico cohort revealed that endogenous *CORT* expression was higher in metastatic PCa samples compared to primary tumors and non-tumor samples (Figure 4D). Indeed, ROC analysis indicated that *CORT* expression significantly discriminated between metastatic vs. non-metastatic samples (AUC = 0.644, *p* = 0.023; Figure 4D).

### 2.5. Endogenous CORT Modulates the Functional and Pharmacological Response of Androgen-Independent PCa Cells

To determine whether the high levels of endogenous CORT found in PCa cells/tissues could exert an autocrine/paracrine regulatory function in AI-PCa cells, we silenced the expression of endogenous *CORT* using a specific and validated siRNA (Figure 5A).

Our results indicate that CORT silencing increased the proliferation rate of 22Rv1 (after 48–72 h) and PC-3 cells (after 24–48–72 h; Figure 5B). However, it should be mentioned that this increase in the proliferative rate seemed to be cell line dependent [i.e., more sustained in time in 22Rv1 (maximum increment after 48–72 h) than in PC-3 cells (maximum increase at 24 h and then, a gradually decrease was observed at 48 and 72 h)], which might be explained in part to a potential different sensitivity to the transient transfection of the two PCa cell lines used (i.e., a loss of function over time is expected in all cell models after a transient transfection), and/or to specific phenotypic differences of the two PCa cell models (i.e., metabolic rate, aggressiveness, etc.). Additionally, no significant changes were observed in the number of colonies formed and in the migration rate in response to endogenous CORT silencing in PC-3 cells (see response to reviewer 1). However, we also explored the phosphorylation levels of AKT and JNK proteins in response to CORT silencing (pathways previously altered in response to CORT peptide treatment) which revealed that levels of JNK were up-regulated only in PC-3 cells (Figure 5C). In addition, gene expression levels of key cell cycle/proliferation markers and SSTRs were also evaluated in response to CORT silencing (Figure 5D). Specifically, a down-regulation in the expression of *CDK2* (in 22Rv1 and PC-3), of *CDKN1B* and *CDKND* (in 22Rv1 cells), and of *CDKN1A* and *CDKN2B* (in PC-3 cells) was observed after CORT silencing (Figure 5D). Interestingly, the silencing of CORT also reduced the expression of *SSTR1*, *SSTR2*, and *SSTR5* in 22Rv1 and the expression of *SSTR5* in PC-3 cells (Figure 5D).

Finally, we also evaluated whether the silencing of endogenous CORT could influence the responsiveness of AI-PCa cells to different SSAs [first generation (octreotide) and second generation (pasireotide)]. Specifically, we found that octreotide and pasireotide significantly reduced proliferation rate in scramble-intact 22Rv1 cells (Figure 5E). Similarly, pasireotide (but not octreotide) also inhibited proliferation rate in scramble-intact PC-3 cells (Figure 5E). In contrast, CORT silencing was able to completely block the antiproliferative effects of octreotide and pasireotide in both AI-PCa cell models (Figure 5E). These results suggest that altered endogenous CORT expression may influence selectively the antitumor response of SSAs in AI-PCa cells.

## 3. Discussion

Despite new advances in clinical practice, the management of PCa remains one of the world’s leading health problems [1,26]. In contrast to localized PCa, advanced disease represents the main cause of PCa-related death, causing more than 350,000 new deaths worldwide per year [1,26,27]. Then, new diagnostic, prognostic, and therapeutic alternatives are urgently needed to improve the clinical management of this pathology. In this sense, it is widely described that some components belonging to the SST/CORT-system are frequently altered and play a critical role in different ERCs, including PCa [18,19,28,29,30]. However, to the best of our knowledge, the pathophysiological role of the two natural ligands belonging to this system, SST and CORT, and their receptors has not been explored in parallel so far in PCa. Therefore, since this system has been very useful in other ERCs to identify new molecular biomarkers to better diagnose, predict prognosis and tumor behavior, and has provided tools to develop novel therapeutic strategies (i.e., SSAs), we aimed to explore the presence of this system (ligands and receptors) and the actions of these peptides and SSAs in PCa cells.

In this work, we observed that the treatment with SST and CORT peptides was able to reduce different key tumor parameters linked to tumor growth and metastasis (i.e., proliferation, migration, and colonies formation) only in AI-PCa cells (22Rv1 and PC-3 cells, two representative models of CRPC pathology), but not in normal prostate and AD-PCa cells, suggesting a potential and specific antitumor capacity of these peptides in the most aggressive phenotype of PCa. Interestingly and in line with these results, our group has recently described that neuronostatin (NST; a recently discovered peptide contained in the preproSST precursor polypeptide encoded by the SST gene but not sharing amino-acid homology to SST) also exerts a specific antitumor capacity in AI-PCa cells [18]. Therefore, all these results might suggest that this complex set of natural ligands (SST, CORT and NST) might exert antitumor actions exclusively in the most aggressive PCa phenotype, which could be considered an important clinical finding as will be discussed below.

Mechanistically, these antitumor effects of SST and CORT were associated with the alteration in the levels of critical genes and oncogenic signaling pathways that have been reported to be frequently associated with the functional and cellular control of the SST system in multiple ERCs (e.g., proliferation, migration, and PCa-aggressiveness features) [14,15,18,20,28,31,32]. Specifically, we found that SST and CORT could exert their antitumor actions in AI-PCa cells through the modulation of AKT, JNK, *MKI67*, *CDK2*, *CDK4*, *CDK6*, *CDKN1A*, *CDKN1B*, *CDKND*, *MMP3*, *MMP9*, *MMP10*, *CDH2*, *EGF*, *EZH2*, *C-MYC*, *PTEN*, and *VEGFR* levels. All these molecular events might be associated with the reduction in the proliferation, migration and colonies formation previously described in response to SST and CORT treatments, wherein some of these changes (especially the alteration of CDK2/4/6 and CDKN1A/1B) might be probably linked to an alteration in the cell cycle arrest (interruption of G1 to S transition), cellular matrix degradation and stem-like cell status [33,34,35,36]. However, it should be mentioned that the modulatory actions of SST and CORT were, in some cases, cell line dependent, which might be explained by the specific phenotypic differences between the two AI-PCA cell models used (i.e., mutation profile, aggressiveness, metabolic rate, etc. [25,37,38]). Additionally, these differences could be also attributed to the differential SSTRs expression profile found between 22Rv1 and PC-3 cells in the present study (i.e., *SSTR1* = *SSTR5* >>> *SSTR2* > *sst5TMD4* > *SSTR3* in 22Rv1 cells vs. *SSTR5* >>> *SSTR2* > *SSTR1* > *sst5TMD4* in PC-3 cells) since it has been reported that each SSTR-subtype can be linked to a different signaling pathway profile [12,39,40]. Nonetheless, our data clearly demonstrate that SST and CORT are functionally active inhibitors of proliferation, migration, and colonies formation exclusively in AI-PCa cells through the modulation of the levels of multiple key signaling molecules related to cancer development, progression and aggressiveness.

In this study, we also had the opportunity to analyze in parallel the expression pattern of all SSTR-subtype by a quantitative PCR method in a representative cohort of PCa tissues, which revealed that *SSTR1*, *SSTR2* and *SSTR5* were highly expressed in PCa tissues (i.e., *SSTR1* ≥ *SSTR2* = *SSTR5*). Our results are in accordance with a previous study from our group indicating that SSTR1 is highly expressed in PCa tissues [18,19,20]. Notably, these data might be considered an important clinical finding because it might suggest that PCa tissues could be sensitive to the actions of SSAs, as the responsiveness of SSAs is critically dependent on the presence of SSTs, and because the treatment with available SSAs [both, first generation (e.g., octreotide) and second generation (i.e., pasireotide)] has become the mainstay of medical therapy for tumor control in different ERCs expressing SSTRs (such as pituitary and gastroenteropancreatic neuroendocrine tumors [22,23,41,42,43]). In fact, we demonstrated that octreotide (which acts primarily by binding to SSTR2 and with less affinity to SSTR5) and pasireotide (a multi-receptor ligand with high affinity for SSTR1, SSTR2, SSTR3, and SSTR5) significantly reduced proliferation rate in 22Rv1 cells (a cell model with an expression profile of *SSTR1* = *SSTR5* >>> *SSTR2*), while only pasireotide, but not octreotide, inhibited proliferation rate in PC-3 (a cell model with an expression profile of *SSTR5*>>>*SSTR2* > *SSTR1*), which reinforce the idea that PCa patients, especially patients with CRPC, could be sensitive to the antitumor actions of SSAs, opening new avenues to explore their potential as targeting therapy for patients with CRPC. Obviously, additional work will be required to evaluate the efficiency of SSAs alone or in combination with other drugs currently used for the treatment of CRPC (i.e., abiraterone or enzalutamide) in patients with CRPC.

Another relevant finding of our study is that we demonstrated that endogenous CORT is highly expressed in 22Rv1 and PC-3 cells, as well as in human PCa tissues compared with BPH samples. Remarkably, we also found that endogenous *CORT* expression was higher in metastatic PCa samples compared to primary tumors and non-tumor samples. As a result, ROC analysis revealed that endogenous *CORT* expression could discriminate between patients with PCa vs. patients with BPH, and also between patients that developed metastasis vs. those that did not. Moreover, we observed that the expression of endogenous CORT was positively correlated with the expression of *SSTR1*, *SSTR2* and *SSTR5* in metastatic PCa tissues but not in non-metastatic tissues. All these results suggest a causal link between dysregulation of endogenous *CORT* expression and PCa progression/aggressiveness and, therefore, that endogenous *CORT* may play a significant autocrine/paracrine pathophysiological role in AI-PCa cells, being its expression functionally linked to the dominant SSTR-subtypes expressed in PCa tissues. This hypothesis was confirmed when we silenced endogenous *CORT* levels in AI-PCa cells which resulted in a significant increase in proliferation rate in these cells, and in the modulation of the expression/levels of critical genes and oncogenic signaling pathways, including the reduction in the expression of different cell cycle inhibitors (e.g., *CDK2*, *CDKN1A*, *CDKN1B*, and/or *CDKND*). Moreover, our data is consistent with previous reports showing that SST, the other main ligand of SSTR-subtypes, also plays an important autocrine/paracrine role in several cellular models including colorectal cancer cells [44,45]. In fact, a constitutive activation of different SSTRs has been also reported since various SSTRs can display a relevant degree of ligand-independent constitutive activity in different cell systems [46]. However, our results have particular relevance because, to the best of our knowledge, this is the first evidence demonstrating a potential autocrine/paracrine regulatory function for endogenous *CORT* in cancer cells, which might be functionally linked to the expression of the dominant receptors expressed in PCa cells (i.e., *SSTR1*, *SSTR2* and *SSTR5*). In support of this idea is the fact that CORT silencing in AI-PCa cells induced significant changes in the expression levels of key cell cycle/proliferation markers and SSTR-subtypes, such as modulation of *CDK2*, *CDKN1A*, *CDKN1B*, *CDKND*, *SSTR1*, *SSTR2*, and/or *SSTR5*.

Remarkably, as previously mentioned, we also demonstrated that the proliferation rate of AI-PCA cells was significantly inhibited in response to octreotide and/or pasireotide in AI-PCa cells; however, when *CORT* expression was silenced, the treatment with these SSAs was completely inefficient in decreasing the proliferation rate, suggesting that the reduction in the levels of *CORT* could desensitize AI-PCa cells to the antitumor actions of SSAs treatment. Interestingly, we found that CORT-silencing was able to significantly down-regulate the expression of the dominant SSTR-subtypes expressed in 22Rv1 (i.e., *SSTR1*, *SSTR2* and *SSTR5*) and in PC-3 (i.e., *SSTR5*), which might in part explain the desensitization observed in CORT-silenced AI-PCA cells to the antiproliferative effects of SSAs. We acknowledge that the limitations of our study are the lack of in vivo preclinical studies analyzing the actions of SST and CORT in the prostate gland physiology under normal and pathological-PCa conditions, the lack of analyzed metastatic CRPC samples in our internal cohort of patients, and that further work will be required to evaluate whether the levels of *CORT* expression could be used as a predictive molecular biomarker to select patients with PCa, especially CRPC, susceptible to being treated with SSAs. Moreover, it seems plausible that additional factors, besides the simple abundance of endogenous *CORT*, might critically influence the SSAs response in PCa cells, including the presence of the truncated splicing sst5TMD4 as has been previously suggested by our group in PCa and other tumor pathologies [14,17,20].

## 4. Materials and Methods

### 4.1. Patients and Samples

This study was approved by the Reina Sofia University Hospital Ethics Committee and conducted in accordance with the ethical standards of the Declaration of Helsinki and the World Medical Association. Core needle biopsies from patients with significant PCa (*n* = 66) and benign prostatic hyperplasia (BPH; *n* = 3; used as control) were collected (cohort-1; results included in Figure 4A-C). The presence or absence of tumor was histologically confirmed by expert uropathologists. Clinical information of patients is provided in Table 1.

The Andalusian Biobank (Córdoba Node) coordinated the collection, processing, management, and assignment of the biological samples used in the present study according to the standard procedures established for this purpose. Written informed consent was obtained from all patients.

In addition, expression levels and clinical data were obtained from the publicly available Grasso cohort [6], which includes metastatic CRPC (*n* = 27), localized prostate adenocarcinomas (*n* = 49), and non-tumor prostate tissue specimens (*n* = 12) (results included in Figure 4D). The data were downloaded from the CANCERTOOL portal [47].

### 4.2. Cell Cultures and Reagents

The normal-like prostate cell line RWPE-1, the Androgen-Dependent (AD) PCa cell model LNCaP, and the two Androgen-Independent (AI) PCa cell models 22Rv1 and PC-3 were obtained from American Type Culture Collection (ATCC, Manassas, VA, USA), and maintained according to manufacturer instructions as previously described [19,20,48]. These cell lines were validated by analysis of short tandem repeats sequences (STRs) using GenePrint 10 System (Promega, Barcelona, Spain), and monthly checked for mycoplasma contamination by polymerase chain reaction (PCR) as previously reported [20]. Human somatostatin-14 (SST-14) and cortistatin-17 (CORT-17) were purchased from Polypeptide Group (Neuhofstrasse, Switzerland). SSAs (octreotide and pasireotide) were obtained from Polypeptide Group and Novartis Pharmaceuticals Corporation, respectively. All these treatments were resuspended in water and used at 10^−7^ M based on previous reports [13,14,49].

### 4.3. Cell Proliferation Assay

As previously described [18,32], cell proliferation was assessed by Resazurin Reagent (# CA035; Canvax Biotech, Córdoba, Spain). Briefly, cells were seeded in 96-well plates at a density of 3000 cells/well, serum-starved overnight, and then fluorescence (540 nm excitation and 590 nm emission) was measured after 3 h incubation with 10% resazurin using the FlexStation III system (Molecular Devices, Sunnyvale, CA, USA). This process was repeated after 24, 48, and 72 h of incubation in response to SST, CORT, octreotide, and pasireotide treatment and/or CORT-silencing (see below) in RWPE-1, LNCaP, 22Rv1, and/or PC-3 cell lines. All the data were normalized to values obtained in day 0 and represented as fold change compared to vehicle-treated controls or scramble-transfected cells.

### 4.4. Cell Migration Assay

Cell migration was evaluated in PC-3 cells, given its high invasiveness nature, as previously reported [50]. Specifically, 30,000 cells were seeded in an Incucyte Imagelock 96-well plate (Cat. No. 4379, Sartorius, Goettingen, Germany). Then, when confluence was reached, cells were starved for 3 h, a scratch was made using Incucyte^®^ Woundmaker Tool (Cat. No. 4563, Sartorius) in each well and the media was replaced by fresh serum-free media. Images of the wound were taken at 0 and after 16 h of incubation with the different treatments. Wound-healing was calculated as the area observed 16 h after the wound was made vs. the area observed just after wounding, using ImageJ software [51].

### 4.5. Colonies Formation

To determine the clonogenic capacity of 22Rv1 and PC-3 PCa cells in response to different treatments, 2000 cells were seeded into 6-well plates, as previously reported [52]. Then, after 10 days, the medium was removed, the colonies washed with PBS, stained with crystal violet solution (crystal violet at 0.05% and glutaraldehyde at 6%) for 30 min, and air-dried. The number of individual colonies was determined by ImageJ software (colony area plugin) [51].

### 4.6. RNA Isolation, Quantitative Real-Time PCR (qPCR), and Customized qPCR Dynamic Array Based on Microfluidic Technology

AllPrep DNA/RNA/Protein Mini Kit (Qiagen, Hilden, Germany) and TRIzol Reagent (Thermo Fisher Scientific, Waltham, MA, USA) were used to isolate RNA from fresh tissues and PCa cell lines, respectively. RNA was DNase-treated using RNase-Free DNase Kit (Qiagen). Total RNA concentration and purity were assessed using Nanodrop One Spectrophotometer (Thermo Fisher Scientific, Madrid, Spain). Total RNA (1 µg) was reverse transcribed using random hexamer primers and the cDNA First-Strand Synthesis kit (Thermo Scientific). Details regarding the development and validation of primers and for the standard real-time qPCR and qPCR microfluidic-based dynamic array technology have been previously reported by our laboratory [53,54]. Detailed information about the primers used herein can be found in Appendix A. To control for variations in the efficiency of the retrotranscription reaction, mRNA copy numbers of the different transcripts analyzed were adjusted by the expression level of a normalization factor (calculated with ACTB and GAPDH expression levels, using GeNorm 3.3) [55].

### 4.7. Western Blotting

22Rv1 and PC-3 cell lines were processed to analyze protein levels by Western-blot after 30 min of SST or CORT exposure or after 48 h of CORT silencing (siRNA transfection) as previously reported by our group [18,53]. Briefly, 300,000 cells were seeded in 6-well plates, and after the experimental procedure described above, proteins were extracted using pre-warmed Sodium Dodecyl Sulfate-Dithiothreitol (SDS-DTT) buffer (62.5 mM Tris-HCl, 2% SDS, 20% glycerol, 100 mM DTT, and 0.005% bromophenol blue). Then, proteins were sonicated for 10 s and boiled for 5 min at 95 °C. Proteins were separated by SDS-PAGE and transferred to nitrocellulose membranes (Millipore, Billerica, MA, USA). Membranes were blocked with 5% non-fat dry milk in Tris-buffered saline/0.05% Tween-20 and incubated overnight with the specific primary antibodies at 1:1000 dilution [phospho-AKT (p-AKT; #4060S; Cell-Signaling, Barcelona, Spain), AKT (#9272S; Cell-Signaling, Barcelona, Spain), phospho-ERK (#4370S; Cell-Signaling), ERK (#9102S; Cell-Signaling), phospho-JNK (#AF1206; RD system), JNK (#AF1387; RD system), phospho-PTEN (#S380; Cell-Signaling), PTEN (9552S; Cell-Signaling), phospho-AR (#16969; Cell-Signaling), AR (ab133273; Abcam)]. Secondary horseradish peroxidase (HRP)-conjugated goat anti-rabbit Immunoglobulin G (IgG) (# 7074S; Cell-Signaling, Barcelona, Spain) or anti-mouse IgG (#7076S; Cell-Signaling) were used at 1:2000 dilution. Proteins were detected using an enhanced chemiluminescence detection system (GEHealthcare, Madrid, Spain) with dyed molecular weight markers (Bio-Rad, Madrid, Spain). Phosphorylation levels of specific proteins were calculated as the ratio between the levels of a specific phospho-protein and its total protein levels detected. Densitometry analysis of the bands obtained was carried out with ImageJ software [51].

### 4.8. Silencing of Endogenous CORT Gene Expression

22Rv1 and PC-3 cells were used for silencing experiments as previously reported [18,20]. Briefly, 300,000 cells were seeded in 6-well plates and grown until 70–90% confluence was reached. Then, cells were transfected (transient transfection) with a specific and validated small-interfering RNA oligo (siRNA) for knockdown of endogenous levels of CORT (#s194341; Thermo Fisher Scientific, Madrid, Spain), along with the SilencerVR Select Negative Control siRNA (#4390843, Thermo Fisher Scientific) at 75 nM, using Lipofectamine-RNAiMAX (#13778-150, Thermo Fisher Scientific), following the manufacturer’s instructions. After 48 h of incubation, cells were collected for validation and seeded to measure proliferation rate.

### 4.9. Statistical Analysis

All the experiments were performed in at least 3 independent experiments (*n* ≥ 3) and with at least 2 technical replicates. Statistical differences between two conditions were calculated by unpaired parametric t-test or nonparametric Mann–Whitney U test, according to normality, assessed by Kolmogorov–Smirnov test. For differences among three conditions, a One-Way ANOVA or Kruskal–Wallis analysis was performed. Spearman’s or Pearson’s bivariate correlations were performed for quantitative variables according to normality. Statistical significance was considered when *p* < 0.05. A trend for significance was considered when p values ranged between > 0.05 and < 0.1. Data represent means ± SEM. All the analyses were assessed using GraphPad Prism 9 (GraphPad 9 Software, La Jolla, CA, USA).

## 5. Conclusions

Taken together, our results unveiled new conceptual and functional avenues in PCa with potential clinical implications, by demonstrating a therapeutic potential of the SST/CORT/SSTRs system and of different SSAs (i.e., octreotide and pasireotide) in AI-PCa cells. Moreover, our results offer original evidence demonstrating that endogenous *CORT* levels are significantly overexpressed in PCa compared with BHP tissues, and in metastatic vs. non-metastatic tissues, and that the modulation of its expression could be a potential therapeutic avenue that should be explored in the future in PCa since its silencing altered the proliferation rates in AI-PCa cells and desensitized these cells to the antitumor effect of octreotide and pasireotide.

## Figures and Tables

**Figure 1 ijms-23-13003-f001:**
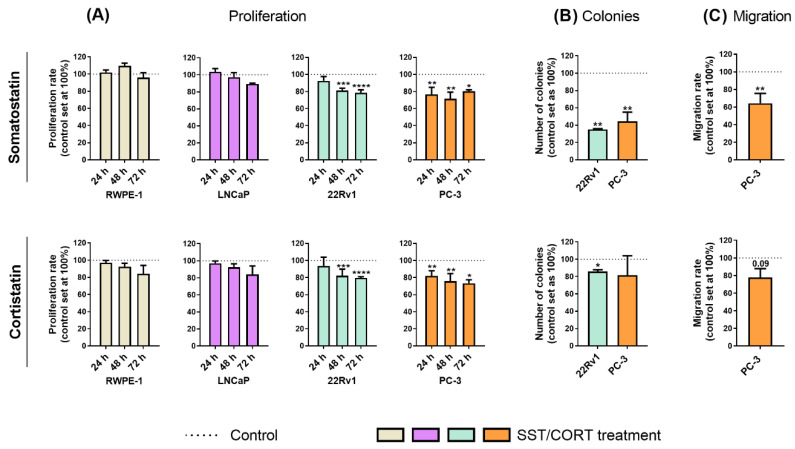
Functional effects after somatostatin (SST) and cortistatin (CORT) treatment in prostate cells. (**A**) Proliferation rate of normal prostate (RWPE-1) and prostate cancer (PCa) cells [androgen-dependent (LNCaP) and androgen-independent (AI; 22Rv1 and PC-3)] in response to SST and CORT treatment (after 24, 48 and 72 h). (**B**) Colonies formation in response to SST and CORT treatment in AI-PCa cells. (**C**) Migration rate of PC-3 cells after 16 h of SST and CORT treatment. Data were represented as percent of vehicle-treated cells (set at 100%). Asterisks (* *p* < 0.05; ** *p* < 0.01; *** *p* < 0.001; **** *p* < 0.0001) indicate statistically significant differences between groups.

**Figure 2 ijms-23-13003-f002:**
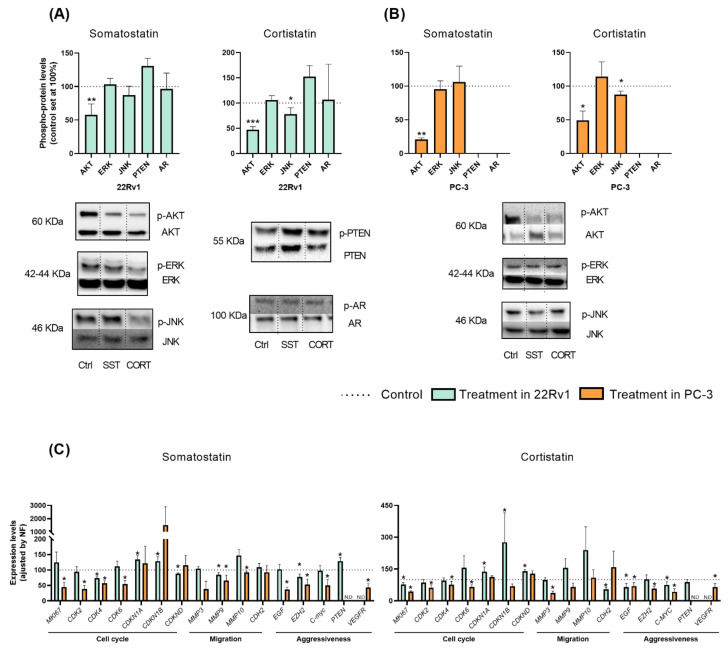
Molecular consequences of somatostatin (SST) or cortistatin (CORT) treatment in androgen-independent (AI) prostate cancer (PCa) cells. (**A**,**B**) Phosphorylation levels of protein belonging to different oncogenic signaling pathways (AKT, ERK, JNK, PTEN and AR) in response to SST and CORT treatment in AI-PCa cells. Phospho-protein levels were normalized by the total amount of each respective protein. Protein data were represented as percent of vehicle-treated cells (set at 100%). (**C**) Fold change in markers of proliferation, migration, and PCa-aggressiveness in response to SST and CORT treatment in AI-PCa cells. Gene expression was represented as the percentage of vehicle-treated cells (set at 100%). Asterisks (* *p* < 0.05; ** *p* < 0.01 and *** *p* < 0.001) indicate statistically significant differences between treatment and vehicle-treated cells. N.D: Non-detected. Ctrl: Control.

**Figure 3 ijms-23-13003-f003:**
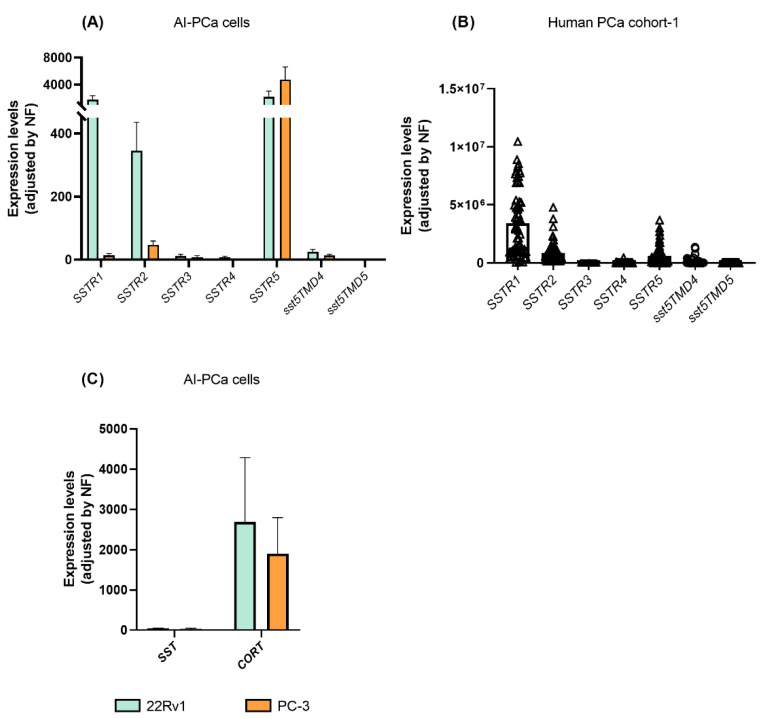
Expression profile of somatostatin-system [receptors (SSTRs), and ligands (somatostatin-SST and cortistatin-CORT)] in androgen-independent (AI) prostate cancer (PCa) cells and fresh prostate tissue (*n* = 69; cohort-1). (**A**) Expression of SSTRs in 22Rv1 and PC-3 AI-PCa cells. (**B**) Expression of SSTRs in PCa fresh samples. (**C**) Expression of SST and CORT in AI-PCa cells. Data represent the mean of mRNA copy number ± Standard Error of the Mean (SEM). mRNA levels were determined by quantitative polymerase chain reaction and adjusted by normalization factor (NF).

**Figure 4 ijms-23-13003-f004:**
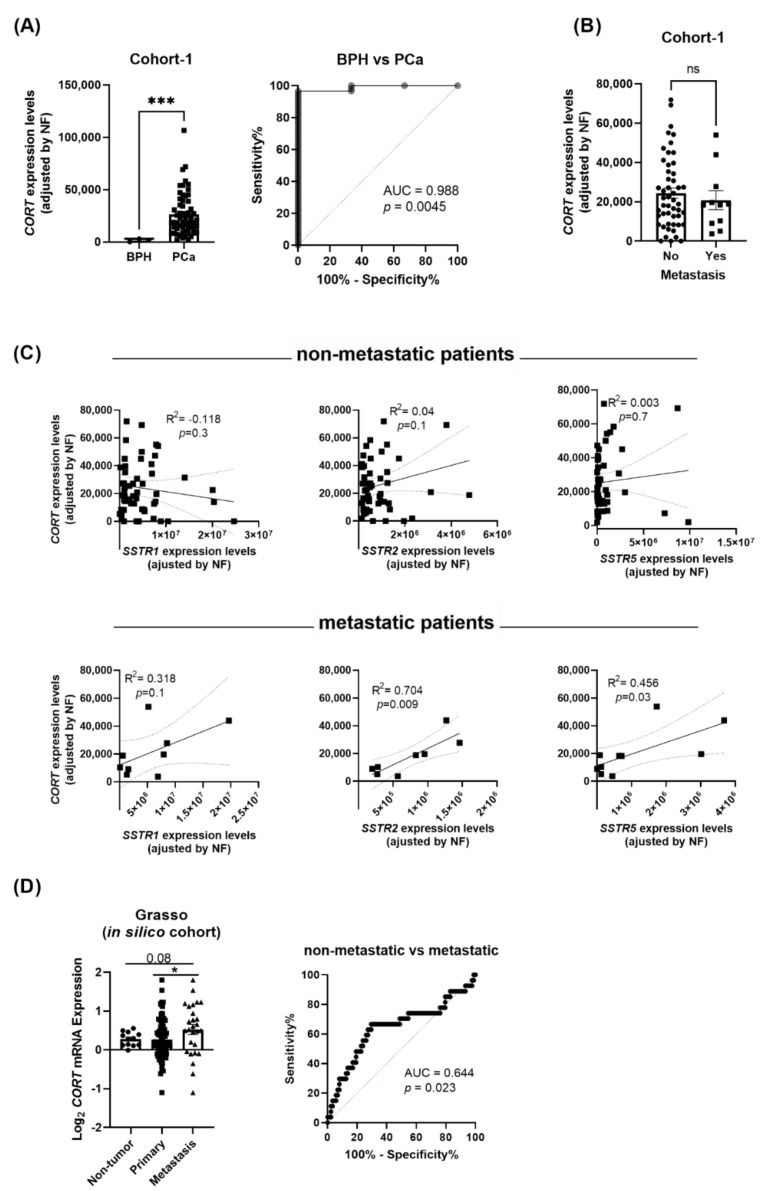
Expression levels of cortistatin (CORT) in prostate tissue. (**A**) Comparison of CORT expression levels between benign prostatic hyperplasia (BPH) and prostate cancer (PCa) samples (cohort-1). ROC curve analysis comparing CORT expression in PCa vs. non-tumor BPH tissues, and associated AUC, is also indicated. (**B**) Comparison of CORT expression between primary tumors obtained from patients with metastasis vs. those without metastasis. (**C**) Correlation between CORT-expression and SSTR1, SSTR2, and SSTR5 expression in primary tumors obtained from patients without and with metastasis (cohort-1). (**D**) Comparation of CORT expression between non-tumor, primary tumor and metastatic samples obtained from the Grasso in silico cohort. ROC curve analysis comparing CORT expression and associated AUC from Grasso cohort is also indicated. mRNA levels were determined by quantitative polymerase chain reaction and adjusted by normalization factor (NF). Asterisks (* *p* < 0.05; and *** *p* < 0.001) indicate statistically significant differences between groups.

**Figure 5 ijms-23-13003-f005:**
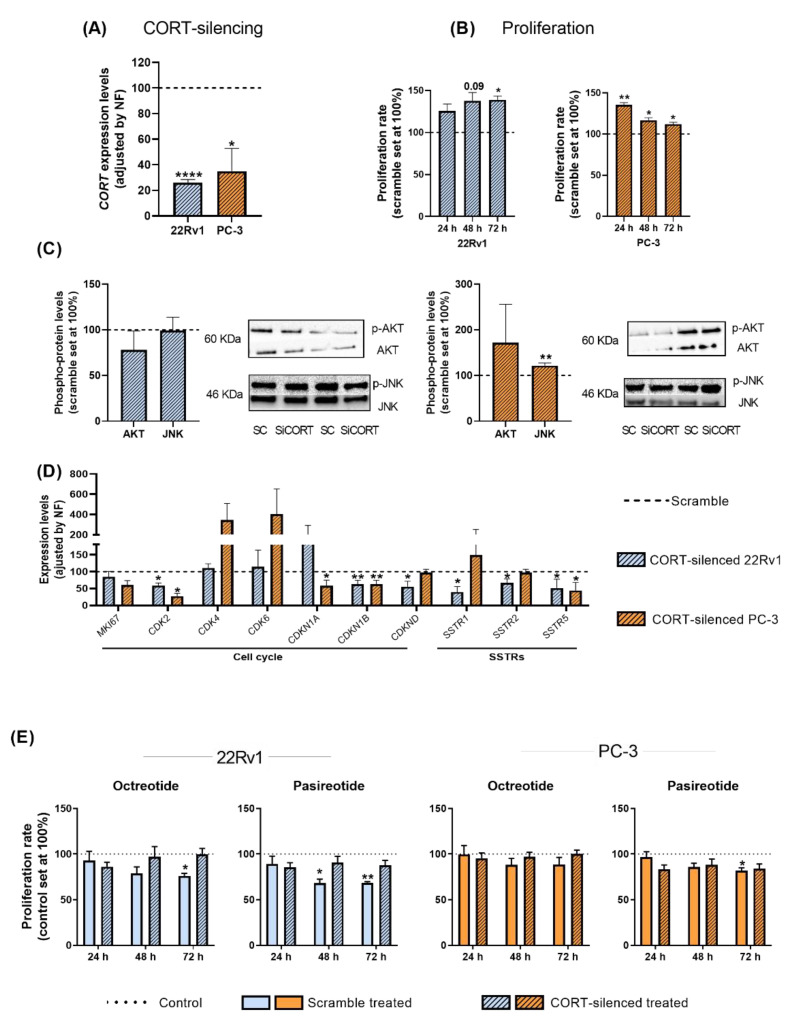
Functional and pharmacological consequences of cortistatin (CORT)-silencing in androgen-independent (AI) prostate cancer (PCa) cells. (**A**) Validation of CORT-silencing in 22Rv1 and PC-3 cells. (**B**) Proliferation rate in response to CORT-silencing in AI-PCa cells. Data were represented as percent of scrambled cells (set at 100%). (**C**) Phosphorylation levels of protein belonging to different oncogenic signaling pathways (AKT, JNK) in response to CORT-silencing in AI-PCa cells. (**D**) Expression of proliferation/cell-cycle and somatostatin receptors genes in response to CORT-silencing in AI-PCa cells. Data were represented as percent of scrambled cells (set at 100%). (**E**) Proliferation rate of scrambled AI-PCa cells or CORT-silenced AI-PCa in response to octreotide and pasireotide. Data were represented as the percent of vehicle-treated cells (set at 100%). Asterisks (* *p* < 0.05; ** *p* < 0.01; **** *p* < 0.0001) indicate statistically significant differences between groups. SC: Scramble. SiCORT: small interference RNA CORT.

**Table 1 ijms-23-13003-t001:** Demographic, biochemical, and clinical parameters of patients with significant PCa. PSA: Prostate-specific antigen.

Patients [*n*]	66
Age, years [median (IQR)]	75 (69–81)
PSA levels, ng/mL [median (IQR)]	62.0 (36.2–254.5)
Gleason score ≥ 7 (%)	66 (100%)
Metastasis (%)	11 (17%)

## Data Availability

The datasets generated and/or analyzed during the current study are available from the corresponding author upon reasonable request.

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
