# Peer review of "Somatostatin, Cortistatin and Their Receptors Exert Antitumor Actions in Androgen-Independent Prostate Cancer Cells: Critical Role of Endogenous Cortistatin"

_ijms, 2022, doi:10.3390/ijms232113003_

Round 1

Reviewer 1 Report

The authors present a manuscript describing the type of somatostatin (SST) and cortistatin (CORT) receptors present in prostate cancer and the effect of SST and CORT on proliferation, clonogenic capacity and migration of prostate cancer cell lines. While certain elements (patient tumor data) of the manuscript is interesting, mechanistic studies presented is limited to cell line studies and is rather preliminary.

Major Issues

·         Do SST and CORT induce cell cycle arrest and/or apoptosis?

·         Does the inhibition of growth and migration seen in vitro also translated to in vivo tumors. There is a lack of preclinical in vivo studies.

·         It is unclear from Fig 4 which data was obtained from patient tumor samples that were collected in this study and which ones were from publicly available data sets.

·         Does CORT silencing affect clonogenic survival and migration too?

·         Does octreotide 231 and pasireotide treatment in CORT silenced cell influence clonogenic capacity or cause cell cycle arrest and/or apoptosis. Importantly, does it translate to preclinical animal xenografts.

Minor comments

Line 44: usually combined with certain drugs such as Abiraterone or~This sentence needs space correction

Line 432: replace retrotranscribed with more appropriate term reverse transcribed.

Reviewer 2 Report

Authors present a study on the role of somatostatin/cortistatin (SST/CORT) system in prostate cancer using cell line models and data from internal and external patient cohorts. The authors use both functional and mechanistic assays to show that SST and CORT have antitumor activity through modulation of oncogenic signaling-pathways and by down-regulating critical genes involved in proliferation/migration and PCa-aggressiveness specifically in androgen-independent prostate cancer cells. The data provided is sufficient to demonstrate the antitumor capacity of SST/CORT and the therapeutic potential of the SST/CORT system in aggressive form of prostate cancer. Few minor revisions are suggested as listed below.

1. Lines 48-50: Authors mention that currently most aggressive form of prostate cancer is treated mainly with androgen-synthesis inhibitors, androgen receptor (AR)-inhibitors, and/or taxanes. A few words could be also mentioned about platinum-based therapies and about PARP-inhibitors as an option for treating aggressive prostate cancer.

2. In figure 2 panel A the cortistatin related bar graph has dashed line instead of dotted line for the control. This should be corrected.

3. In figure 3 panel B the authors present levels of SSTR-subtypes in patient samples, however, it is not clear for the reader to which patient cohort authors are referring. This should be clarified at least in the associated text section (lines 167-168). Also, it would be good to include the number of samples in the figure legend.

4. In lines 235-236 authors note that silencing of endogenous CORT in PC-3 cells increases the proliferation rate. Overall, the proliferation is indeed increased in comparison to control but the proliferation of the CORT silenced PC-3 cells also appears to decrease from the 24h timepoint to the 72h one. Could the authors briefly comment potential reasons why this occurs as same is not observed in 22Rv1 cells?

Round 2

Reviewer 1 Report

Although this reviewer feels the manuscript can be enriched by in vivo data, given the time needed to generate this data, and considering merits of the present data, this reviewer feels that the manuscript can be accepted in its present form. Specially after the authors have made it clear that their data has limitations and also provided additional data (only for review purpose) satisfactorily addressing the concerns of this reviewer.